# Diagnostic Value of Anti-HTLV-1-Antibody Quantification in Cerebrospinal Fluid for HTLV-1-Associated Myelopathy

**DOI:** 10.3390/v16101581

**Published:** 2024-10-08

**Authors:** Tomoo Sato, Naoko Yagishita, Natsumi Araya, Makoto Nakashima, Erika Horibe, Katsunori Takahashi, Yasuo Kunitomo, Yukino Nawa, Isao Hamaguchi, Yoshihisa Yamano

**Affiliations:** 1Department of Rare Diseases Research, Institute of Medical Science, St. Marianna University School of Medicine, Kawasaki 216-8512, Japan; tomoo@marianna-u.ac.jp (T.S.); yagi@marianna-u.ac.jp (N.Y.); araya@marianna-u.ac.jp (N.A.); makoto.nakashima@marianna-u.ac.jp (M.N.); e-horibe@marianna-u.ac.jp (E.H.); takahashi@marianna-u.ac.jp (K.T.); y-kunitomo@marianna-u.ac.jp (Y.K.); 2Department of Neurology, St. Marianna University School of Medicine, Kawasaki 216-8511, Japan; 3Institute of Radioisotope Research, St. Marianna University Graduate School of Medicine, Kawasaki 216-8512, Japan; yukino@marianna-u.ac.jp; 4Research Center for Biological Products in the Next Generation, National Institute of Infectious Diseases, Tokyo 208-0011, Japan; i.hamaguchi.05910@ota-hosp.or.jp; 5Department of Clinical Laboratory, Subaru Health Insurance Society Ota Memorial Hospital, Ota 373-8585, Japan

**Keywords:** human T-cell leukemia virus type 1, HTLV-1-associated myelopathy, diagnosis, cerebrospinal fluid, anti-HTLV-1 antibody

## Abstract

The diagnostic accuracy of cerebrospinal fluid (CSF) anti-human T-cell leukemia virus type I (HTLV-1) antibody testing for HTLV-1-associated myelopathy/tropical spastic paraparesis (HAM) remains unclear. Therefore, we measured the anti-HTLV-1 antibody levels in CSF using various test kits, evaluated the stability of CSF antibodies, and performed a correlation analysis using the particle agglutination (PA) method, as well as a receiver operating characteristic (ROC) analysis between patients with HAM and carriers. The CSF anti-HTLV-1 antibody levels were influenced by freeze–thaw cycles but remained stable when the CSF was refrigerated at 4 °C for up to 48 h. Measurements from 92 patients (69 patients with HAM and 23 carriers) demonstrated a strong correlation (r > 0.9) with the PA method across all six quantifiable test kits. All six test kits, along with CSF neopterin and CXCL10, exhibited areas under the ROC curve greater than 0.9, indicating a high diagnostic performance for HAM. Among these, five test kits, Lumipulse and Lumipulse Presto HTLV-I/II, HISCL-UD (a kit under development), HTLV-Abbott, and Elecsys HTLV-I/II, established a cutoff with 100% sensitivity and maximum specificity, achieving a sensitivity of 100% and a specificity ranging from 43.5% to 56.5%. This cutoff value, in combination with clinical findings, will aid in the accurate diagnosis of HAM.

## 1. Introduction

Human T-cell leukemia virus type I (HTLV-1) is a retrovirus that causes adult T-cell leukemia/lymphoma (ATL) and HTLV-1-associated myelopathy/tropical paraparesis (HAM). HAM is an inflammatory neurological disease that develops insidiously [1,2,3,4]. Gait disturbances due to spastic paraparesis in HTLV-1-infected individuals should prompt a suspicion of HAM. However, diagnosing HAM is challenging and requires differentiation from other conditions such as multiple sclerosis, neuromyelitis optica spectrum disorder, other forms of myelitis, hereditary spastic paraplegia, compressive myelopathy, spinal cord tumors, subacute combined degeneration, and spinocerebellar ataxia [5,6,7].

In Japan, an anti-HTLV-1 antibody titer of 4× or more in cerebrospinal fluid (CSF), as measured by the gelatin particle agglutination (PA) method [8], has been used as a diagnostic cutoff for HAM [9,10]. However, the rationale for this cutoff is largely empirical, and the accuracy in differentiating HAM from HTLV-1 carriers remains uncertain. Furthermore, since the PA-based test kit is no longer available in Japan, it is essential to establish cutoff values for anti-HTLV-1 antibody test kits based on alternative measurement principles.

In addition to the PA method, other anti-HTLV-1 antibody testing methods used for screening in Japan include the chemiluminescent enzyme immunoassay (CLEIA) [11], the chemiluminescent immunoassay (CLIA) [12,13], the electrochemiluminescent immunoassay (ECLIA) [14], and immunochromatography (IC) [15]. The line immunoassay (LIA) [16] is employed for confirmatory testing. The CLEIA, CLIA, and ECLIA quantitatively measure antibody concentrations, with a signal intensity that is proportional to antibody levels. In contrast, IC and the LIA are visually assessed and do not provide quantitative measurements. These assays were primarily developed for serum or plasma samples, and little is known about their applicability to CSF. In particular, the stability of anti-HTLV-1 antibodies in CSF has not been thoroughly investigated, including the impact of freeze–thaw cycles and refrigerated storage time.

Moreover, as all these tests are currently approved as qualitative tests, their quantitative potential has yet to be fully validated. Specifically, it remains to be determined whether the quantitative results of these methods correlate with the antibody titers obtained from the PA method. 

Therefore, in this study, we utilized multiple test kits based on various measurement principles available in Japan to (1) evaluate the stability of anti-HTLV-1 antibodies in CSF (considering the effects of freeze–thaw cycles and refrigerated storage at 4 °C) and (2) assess their diagnostic performance for HAM, establish appropriate cutoff values, and determine correlations with the PA method.

## 2. Materials and Methods

### 2.1. Research Framework and Implementation Plan

This study was conducted in cooperation with four companies (Fujirebio, Tokyo, Japan; Sysmex, Kobe, Japan; Abbott Japan, Tokyo, Japan, and Roche Diagnostics, Tokyo, Japan), all of which have developed anti-HTLV-1 antibody test kits. Clinical specimens were sent to each company, and assays were performed using their respective kits. The measurement results were returned to us, and we conducted analyses independently.

### 2.2. Anti-HTLV-1 Antibody Test Kits

The anti-HTLV-1 antibody test kits used in this study are listed in Table 1. One of the test kits is currently under development (UD). Two types of assays were used: a sandwich assay, where anti-HTLV-1 antibodies are sandwiched between HTLV-1 antigens, and an indirect assay, which detects anti-HTLV-1 antibodies using anti-human IgG antibodies.

### 2.3. Ethical Considerations

All subjects provided written informed consent for HAM research prior to the collection of CSF and blood samples using a consent form approved by the Bioethics Committee of the St. Marianna University School of Medicine (Approval No. 1646). As this was a retrospective study using stored CSF samples along with clinical and laboratory data, the requirement for informed consent from each patient was waived by the Bioethics Committee (Approval ID No. 5829). The patients were guaranteed the opportunity to opt out of participation following information disclosure on the study website (https://nanchiken.jp/saihatsu, accessed on 6 October 2024).

### 2.4. Subjects, Specimen Sampling, and Laboratory Analysis

The HTLV-1 infection status was determined based on screening and confirmation tests, which are standard methods in Japan [17]. Enrolled patients with HAM were diagnosed according to the WHO diagnostic criteria [18], and those not diagnosed were defined as HTLV-1 carriers. Peripheral blood mononuclear cells (PBMCs) were isolated from peripheral blood using density gradient centrifugation over Pancoll human (PAN-biotech, Aidenbach, Germany). Genomic DNA was extracted from PBMCs using a FlexiGene DNA Kit (QIAGEN K.K., Tokyo, Japan). The CSF specimens were obtained via lumbar puncture. A portion of each CSF sample was used for laboratory tests, including those for anti-HTLV-1 antibody titers (PA method), total protein levels, cell counts, and inflammatory markers (CXCL10 and neopterin). The CSF CXCL10 concentrations were measured using a cytometric bead array (BD Biosciences, Franklin Lakes, NJ, USA), and the CSF neopterin concentrations were measured using high-performance liquid chromatography (HPLC) at SRL Inc. (Tokyo, Japan). The remaining CSF specimens were centrifuged, and the cell pellets were lysed in a buffer containing proteinase K. To measure the proviral load in CSF cells, genomic DNA was extracted using the phenol–chloroform method. The HTLV-1 proviral load was measured using genomic DNA from PBMCs and CSF cells as templates, following a previously reported method [19]. The CSF supernatant was aliquoted into cryotubes and stored at −80 °C for future use.

### 2.5. Assessment of CSF Anti-HTLV-1 Antibody Stability

To evaluate the stability of anti-HTLV-1 antibodies in the CSF, we selected stored CSF specimens from 12 patients with HAM, including four low-titer (16×/32× by the PA method), four medium-titer (128×/256×), and four high-titer (512×/1024×) samples. The samples were collected between 2014 and 2022. These 12 CSF specimens were thawed, aliquoted, and refrozen. Six specimens were sent to each company (two low-titer, two medium-titer, and two high-titer samples per company). 

Each company performed the assay using their proprietary anti-HTLV-1 antibody test kit under the following conditions: (1) after one, two, and three freeze–thaw cycles and (2) after 0 h, 24 h, and 48 h of refrigerated storage at 4 °C.

### 2.6. Distribution Study of CSF Anti-HTLV-1 Antibody Titers in Patients with HAM

For the distribution study, we used data on initial CSF anti-HTLV-1 antibody titers (PA method) and the presence or absence of steroid therapy at the time of CSF collection from 322 patients with HAM, obtained between 2007 and 2022.

### 2.7. Assessment of Diagnostic Performance of Anti-HTLV-1 Antibody Test Kits

We evaluated the diagnostic performance and cutoff values of each anti-HTLV-1 antibody test kit using 92 CSF specimens from patients with HAM (n = 69) and HTLV-1 carriers (n = 23). HTLV-1 carrier CSF samples were obtained from 23 patients in whom CSF testing was performed between 2013 and 2022, with no diagnosis of HAM. The number of HAM patient-derived CSF specimens was set to 69, which was thrice the number of HTLV-1 carriers. The 69 CSF samples from patients with HAM were chosen to reflect the same distribution of anti-HTLV-1 antibody titers as the steroid-naive group described in Section 2.6, based on the PA method data. The 92 CSF specimens were thawed once, aliquoted, refrozen, and sent to each company for analysis using anti-HTLV-1 antibody test kits. 

Additionally, we collected data on sex, age, HTLV-1 proviral load, CSF CXCL10 concentration, and CSF neopterin concentration for all 92 patients.

### 2.8. Statistical Analysis

Anti-HTLV-1 antibody quantitative data (COI and S/CO) and antibody titers measured using the PA method were log_10_- or log_2_-transformed, respectively, to approximate a normal distribution. For values of zero, 0.01 was substituted to enable logarithmic conversion. A one-way repeated-measures ANOVA was used to compare data from the three conditions in Section 2.5. For comparisons between the two groups of HTLV-1 carriers and patients with HAM concerning sex, Fisher’s exact test was used. The Mann–Whitney U test was used for all comparisons except for those concerning sex. A receiver operating characteristic (ROC) curve analysis was performed to evaluate the diagnostic performance of several known markers and anti-HTLV-1 antibody test kits for HAM diagnosis. The overall concordance rate between the PA method and each other test kit was calculated when “a 4× or greater PA method antibody titer” or “a 16× or greater antibody titer” was considered positive. Spearman’s rank correlation analysis was performed to examine the correlation between the PA method and each test kit. In all cases, *p* ≤ 0.05 was considered statistically significant. Statistical analyses and graph compositions were performed using Prism 8 software (GraphPad Software Inc., San Diego, CA, USA).

## 3. Results

### 3.1. Assessment of Stability of CSF Anti-HTLV-1 Antibody Measurement

The PA method (SERO) showed a trend toward increased antibody titers following freeze–thaw cycles. However, this difference was not statistically significant (*p* = 0.0756; Figure 1A). In contrast, for the other methods (Figure 1B–F), the antibody levels (COI and S/CO) after the third freeze–thaw cycle were lower than those after the first freeze–thaw cycle. For three of the test kits (LU, LU-P, and Abbott), the antibody levels decreased significantly with each freeze–thaw cycle (Figure 1B,C,E). Regarding the storage time at 4 °C, the PA method (SERO) showed an increasing trend in the antibody titer as the storage time increased from 0 to 48 h (Figure 2A). However, the changes in antibody levels over the 48-h storage period in all methods, including the PA method, were not significant (Figure 2B–F).

### 3.2. Assessment of Diagnostic Performance of Anti-HTLV-1 Antibody Test Kits

#### 3.2.1. Distribution of CSF Anti-HTLV-1 Antibody Titers Using the PA Method in Patients with HAM

First, we analyzed the distribution of CSF anti-HTLV-1 antibody titers (PA method) in patients with HAM (Figure 3). We collected data from 322 patients with HAM, including 248 who were steroid-free and 74 who received steroid treatment. In the steroid-free group, the median CSF anti-HTLV-1 antibody titer was 128× (range: 4×–8192×), with a mean ± SD of 7.0 ± 2.4 on a log_2_ scale. In the steroid-treated group, the median titer was also 128× (range: 4×–4096×), with a mean ± SD of 7.1 ± 2.2 (log_2_). There was no statistically significant difference in the distribution of CSF antibody titers between the two groups (*p* = 0.627). More than 80% of patients with HAM, regardless of steroid treatment, had antibody titers in the range of 16× to 512× (82% untreated, 84% treated).

#### 3.2.2. Characteristics of HTLV-1 Carriers and Patients with HAM Used to Determine Diagnostic Performance

We prepared 92 CSF specimens from 23 HTLV-1 carriers and 69 patients with HAM to assess the diagnostic performance of the anti-HTLV-1 antibody test kits. The 69 samples from patients with HAM were selected to match the distribution of anti-HTLV-1 antibody titers in the steroid-free group, as described above. The characteristics of the patients with HAM and HTLV-1 carriers are summarized in Table 2. There were no significant differences in terms of sex between the groups. However, significant differences were observed in terms of age, the HTLV-1 proviral load (PVL) in PBMCs and CSF cells, the ratio of the PVL in CSF cells to that in PBMCs, and CSF inflammatory markers (CSF CXCL10 and CSF neopterin levels). These findings are consistent with those of previous studies [20,21,22,23]. Notably, the CSF inflammatory markers had an AUC greater than 0.9, demonstrating strong diagnostic performance in distinguishing between patients with HAM and HTLV-1 carriers (Figure 4). In contrast, the PVL in CSF cells, the ratio of the PVL in CSF cells to the PVL in PBMCs, and the PVL in PBMCs had lower AUC values. Among the 23 HTLV-1 carriers, 2 of 20 (10%) had an undetectable PVL in their CSF cells, excluding 3 cases in which PVL data were unavailable.

#### 3.2.3. Diagnostic Performance of Anti-HTLV-1 Antibody Test Kits

To evaluate the diagnostic performance of the seven quantifiable test kits (excluding the non-quantifiable IC_ESPLINE and LIA_INNO-LIA) in distinguishing patients with HAM from HTLV-1 carriers, an ROC analysis was performed using data from 92 cases (Figure 5). All seven test kits demonstrated AUCs greater than 0.9, indicating excellent diagnostic performance. Considering the 95% confidence interval, these seven kits were considered to have approximately equivalent diagnostic performance.

#### 3.2.4. Evaluation of the 4× Cutoff Value for the PA Method

Next, we examined the appropriateness of the 4× cutoff value for the PA method, which has been used as a cutoff for diagnosing HAM in Japan. We assessed the overall concordance rate between the PA method with a 4× or 16× antibody titer as the cutoff value and each test kit. For example, when the PA method 4× was used as the cutoff, the CLEIA_LU test achieved an overall concordance rate of 94.6% (Table 3). However, when the PA 16× cutoff was applied, the overall concordance rate dropped to 81.5%, primarily because of an increase in false positives (Table 4). Similarly, we calculated the overall concordance rate for all test kits using PA cutoff values of 4× and 16× (Table 5). HISCL and INNO-LIA, which use anti-human IgG antibodies as secondary reaction reagents, showed higher false-negative rates and lower overall concordance rates with the PA 4× cutoff than with the 16× cutoff. Conversely, test kits based on the sandwich assay method had higher overall concordance rates with the PA 4× cutoff, owing to fewer false positives (Table 5). These findings suggest that a PA cutoff of 4× is appropriate for use with CSF. However, with this cutoff, the sensitivity and specificity for detecting HAM were 100% and 30.4%, respectively, indicating low specificity (middle part of Table 6).

#### 3.2.5. Cutoff Values for Diagnosing HAM

Five quantifiable anti-HTLV-1 antibody test kits (LU, LU-P, UD, Abbott, and Elecsys), all based on the sandwich method, achieved 100% sensitivity using the same criteria as the blood test (COI > 1.0, S/CO > 1.0) as the cutoff values. However, the specificity was low, ranging from 8.7% to 34.8% (upper part of Table 6). Therefore, we set a cutoff higher than the COI (1.0) or S/CO (1.0) to improve the specificity without compromising the sensitivity. 

Consequently, to identify appropriate cutoff values to distinguish patients with HAM from carriers, we investigated the correlations of the six quantifiable anti-HTLV-1 antibody tests with the PA method and determined a cutoff corresponding to the PA method 4× (Figure 6). The sensitivity and specificity of the newly determined cutoff values were also calculated (middle part of Table 6). All the test kits showed significant correlations with the PA method, with Spearman’s rank correlation coefficients greater than 0.9. Although using cutoff values equivalent to the PA method 4× increased the specificity, three test kits had sensitivities below 100%, indicating that some HAM cases could be missed.

Therefore, we established more appropriate cutoff values that maximized specificity while maintaining 100% sensitivity (lower part of Table 6). Using these cutoffs, we achieved a sensitivity of 100% and specificity of 43.5–56.5% for all five quantifiable test kits except SERO and HISCL.

#### 3.2.6. Diagnostic Performance of IC_ESPLINE and LIA_INNO-LIA

Finally, we examined whether the same criteria for IC_ESPLINE and LIA_INNO-LIA as those used for blood applied to the diagnosis of HAM. The results showed that IC had the same sensitivity and specificity as the PA method 4× (sensitivity, 100%; specificity, 30.4%). The LIA showed a sensitivity of 91.3% and a specificity of 65.2%, being higher in specificity but lower in sensitivity than the quantifiable test kits (lower part of Table 6).

## 4. Discussion

All the test kits used in this study were able to measure anti-HTLV-1 antibodies in CSF. Among these, six quantitative anti-HTLV-1 antibody test kits (LU, LU-P, HISCL, UD, Abbott, and Elecsys) demonstrated a strong correlation (rs > 0.9) with the PA method (Figure 6), indicating their ability to quantify anti-HTLV-1 antibodies in the CSF. These kits also exhibited high diagnostic performance for HAM, with ROC-AUC values exceeding 0.9 (Figure 4; ROC-AUC 0.927–0.944). Based on the ROC analysis, we determined helpful cutoff values that could serve as alternatives to the PA method 4× for diagnosing HAM (lower part of Table 6).

To date, various methods have been used to detect anti-HTLV-1 antibodies in the CSF of patients with HAM and HTLV-1 carriers. First, it was reported that PA (Serodia-ATLA) and recombinant gag–env hybrid protein-coated ELISAs are useful for detecting anti-HTLV-1 antibodies in CSF and diagnosing HAM [24]. Western blotting has been used to detect GD21, rgp46-I, and p24 as targets of CSF anti-HTLV-1 antibodies in many patients with HAM [25]. In addition, an EIA using 16 synthetic peptides derived from HTLV-1 gag and env was used to measure the peptide-specific IgG antibody levels. Patients with HAM show an intrathecal immune response to more gag and env epitopes than HTLV-1 carriers do [26]. However, in these studies, the accuracy of HAM diagnosis was not examined. In contrast, a study by Kodama et al. showed the diagnostic accuracy of CSF anti-HTLV-1 antibodies using the CLIA and CLEIA methods [10]. However, it showed the cutoff and diagnostic accuracy when the PA results were considered to be true and did not show the accuracy of diagnosing HAM clinically. In this context, this study clarified the accuracy of HAM diagnosis using additional test kits.

We evaluated two aspects of CSF anti-HTLV-1 antibody stability that had yet to be clarified. First, after refrigerated storage at 4 °C, the PA method showed an increase in the antibody titer, whereas the other test kits showed stable antibody levels for at least 48 h (Figure 2). Second, regarding freeze–thaw cycles, all methods showed significant changes or trends toward changes in antibody values (Figure 1). Freeze–thawing does not significantly affect the antibody levels of some viruses in serum [27]. However, the protein concentration in CSF is less than 1/200 that in serum, making it more susceptible to protein denaturation during freezing and thawing [28,29]. Therefore, freezing and thawing of CSF should be avoided. Considering feasibility and reproducibility, we recommend setting the number of freeze–thaw cycles to one and using the same conditions each time.

This study also revealed the distribution of CSF anti-HTLV-1 antibody titers (PA method) among patients with HAM (Figure 3). The mean ± SD of the log_2_ antibody titer in the steroid-free group was 7.0 ± 2.4, consistent with the CSF anti-HTLV-1 antibody titers in untreated patients with HAM (mean log_2_ 6.7 ± 2.5) previously reported by Kodama et al. [10]. 

Our findings suggest that the PA method 4× cutoff is more appropriate for diagnosing HAM than the serum-based 16× cutoff. The reason we considered was that the overall concordance rates between the PA method and each sandwich method test kit (LU, LU-P, UD, Abbott, Elecsys, or ESPLINE) were higher when using the 4× cutoff compared to the 16× cutoff (Table 5). Similarly to the PA method, the sandwich method test kits detected both IgG and IgM anti-HTLV-1 antibodies. In contrast, the test kits utilizing the indirect method (CLEIA_HISCL and LIA_INNO-LIA), which detect only IgG antibodies, had higher rates of false negatives and lower overall concordance rates than the PA method 4× as a standard. This may be attributable to the presence of anti-HTLV-1 IgM antibodies in the CSF of patients [30]. However, it is important to note that while the PA method 4× cutoff is useful, when used to diagnose HAM, it had a relatively low specificity of 30.4% (Table 6). This limitation was highlighted in a previous study [31], and this study provides further evidence of this issue. In addition, because the PA method involves a visual determination of gelatin agglutination, there is a problem in that the interpretation of the results is more prone to variability than it is with other immunoassay kits that automatically measure the signal intensity of chemiluminescence or electrochemiluminescence.

There are some well-known methods for determining optimal cutoff values that balance sensitivity and specificity, such as the maximum Youden index and minimum distance to the upper-left corner of the ROC plot [32]. Kodama et al. employed the maximum Youden index to determine the cutoff value for the diagnosis of HAM [10]. However, the cutoff value can be determined based on its purpose. In this study, the aim of determining the cutoff value was to detect patients with HAM without missing any cases while excluding as many non-HAM HTLV-1 carriers as possible. Therefore, a cutoff value that achieves 100% sensitivity while maximizing specificity is ideal. Based on this approach, we established optimal cutoff values (lower part of Table 6). Using these cutoff values, five test kits (LU, LU-P, UD, Abbott, and Elecsys) achieved higher specificity (43.5–56.5%) than the PA method (30.4%) while maintaining 100% sensitivity, making them more effective for HAM diagnosis. The cutoff values determined in this study were higher than the serum cutoff values (COI 1.0, S/CO 1.0), and anti-HTLV-1 antibodies were detected even in HTLV-1 carriers. This suggests that differentiating patients with HAM from carriers is based not on the mere presence of antibodies in the CSF but rather on the quantity of antibodies. If all cases with COI > 1.0 or S/CO > 1.0 were diagnosed with HAM, there would be a risk of misdiagnosing many carriers as having HAM.

We also assessed the diagnostic performance of two non-quantitative anti-HTLV-1 antibody assays, IC_ESPLINE and LIA_INNO-LIA. These qualitative assays had fixed cutoffs, and their sensitivities and specificities were determined. In contrast, quantitative tests allow the cutoff point to be set freely, and the sensitivity and specificity can be changed, although there is a trade-off between them. Although we wanted to prioritize the sensitivity and set it at 100%, the sensitivity of the INNO-LIA was as low as 91.3%. Next, when the cutoff value of the quantitative test kits was adjusted to achieve a sensitivity of 91.3%, their specificities ranged from 73.9% to 84.1%, which exceeded the specificity of the INNO-LIA (65.2%). This indicates that the quantitative tests demonstrated superior discriminatory ability. The other test, ESPLINA, achieved a target sensitivity of 100%; however, its specificity was low (30.4%) and inferior to that of the quantitative test kits (43.5–56.5%). Therefore, quantifiable test kits should be prioritized over qualitative assays for HAM diagnosis.

In addition to anti-HTLV-1 antibodies, the CSF inflammatory markers CXCL10 and neopterin were effective in distinguishing patients with HAM from HTLV-1 carriers (Figure 4; AUCs of 0.927 and 0.942, respectively), which is consistent with previous findings [23]. Conversely, the diagnostic accuracies of the PVL in CSF cells and the ratio of the PVL in CSF cells to that in PBMCs were relatively low (AUCs of 0.8111 and 0.719, respectively). These findings differ slightly from those of previous reports suggesting that these markers are useful for HAM diagnosis [21]. These discrepancies may be due to differences in the study populations. Regarding other CSF markers, it is known that there is no difference between patients with HAM and HTLV-1 carriers in terms of IgG and total protein in the CSF and that the ability of the CSF cell count to distinguish HAM is lower than that of CSF CXCL10 and neopterin [23]. In this study, patients with HAM had higher levels of CSF inflammatory markers associated with Th1 responses and higher CSF anti-HTLV-1 antibody titers than HTLV-1 carriers, indicating that both cellular and humoral immunity play a role in HAM pathogenesis. This finding is supported by previous studies [5,33].

There were limitations to this study:The CSF specimens used to determine the cutoff values in this study were frozen and thawed twice and underwent a long storage time (5.2 ± 2.2 years) at −80 °C.The CSF specimens were collected exclusively from Japanese patients, and the sample size was relatively small.Some HTLV-1 carriers in this study exhibited neurological symptoms, although HAM was ruled out.

Considering these points, the appropriate cutoff values may vary depending on the conditions of the specimens and characteristics of the target population. Therefore, the cutoff values presented here should be used only as a guide, and the diagnosis of HAM should always be made in conjunction with clinical findings.

## 5. Conclusions

This study established appropriate cutoff values for diagnosing HAM using six quantifiable antibody test kits as an alternative to the PA method 4×. Although this study had limitations, the cutoff values, in combination with clinical findings, could contribute to a more accurate diagnosis of HAM. Future studies are needed to identify novel markers with higher sensitivity and specificity for HAM.

## Figures and Tables

**Figure 1 viruses-16-01581-f001:**
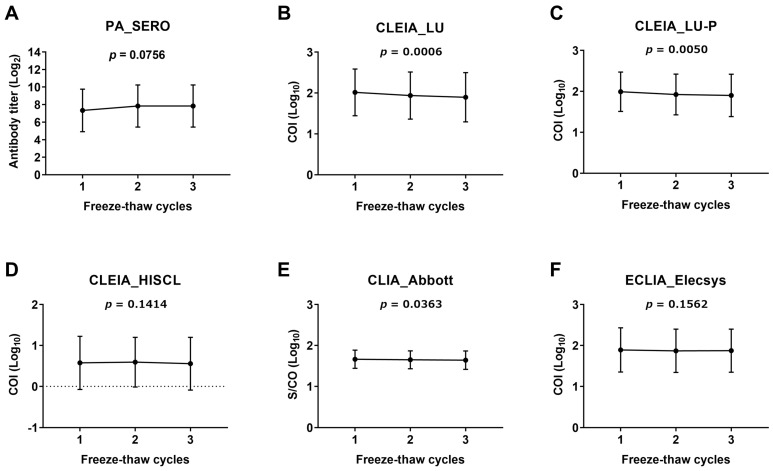
Effect of freeze–thaw cycles on the measurement of anti-HTLV-1 antibody quantitative data in CSF. Six cases of CSF—two low, two medium, and two high titers—were employed in the assay for each test kit (see the Section 2.5 for details). The CSF underwent one, two, or three freeze–thawing cycles, and the anti-HTLV-1 antibody titers were assessed. Measurements for each time point are displayed as the mean ± SD. A one-way repeated-measures ANOVA was used to compare the corresponding data among the three conditions. *p* ≤ 0.05 was considered statistically significant. COI, cutoff index; S/CO, signal-to-cutoff ratio. Information on each test kit is shown in Table 1.

**Figure 2 viruses-16-01581-f002:**
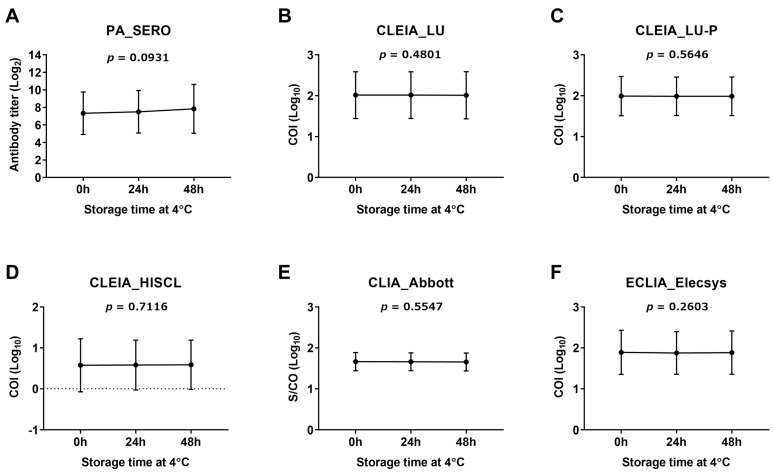
Effect of storage time at 4 °C on the measurement of anti-HTLV-1 antibody quantitative data in CSF. Six cases of CSF—two low, two medium, and two high titers—were employed in the assay for each test kit (see the Section 2.5 for details). The CSF was stored at 4 °C for 0 h, 24 h, and 48 h, and the anti-HTLV-1 antibody titers were assessed. Measurements for each time point are displayed as the mean ± SD. A one-way repeated-measures ANOVA was used to compare the corresponding data among the three conditions. *p* ≤ 0.05 was considered statistically significant. COI, cutoff index; S/CO, signal-to-cutoff ratio. Information on each test kit is shown in Table 1.

**Figure 3 viruses-16-01581-f003:**
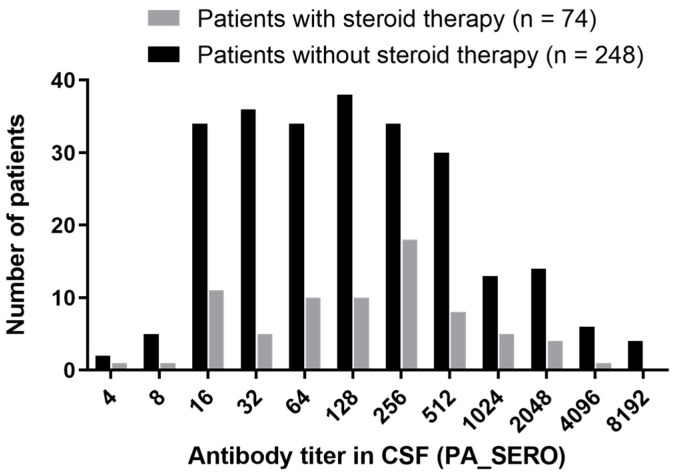
Distribution of CSF anti-HTLV-1 antibody titers in patients with HAM. The vertical axis represents the number of patients, and the horizontal axis represents the antibody titer measured using the PA method. Of the 322 patients with HAM, 248 were not on steroid therapy (black), and 74 were on steroid therapy (gray).

**Figure 4 viruses-16-01581-f004:**
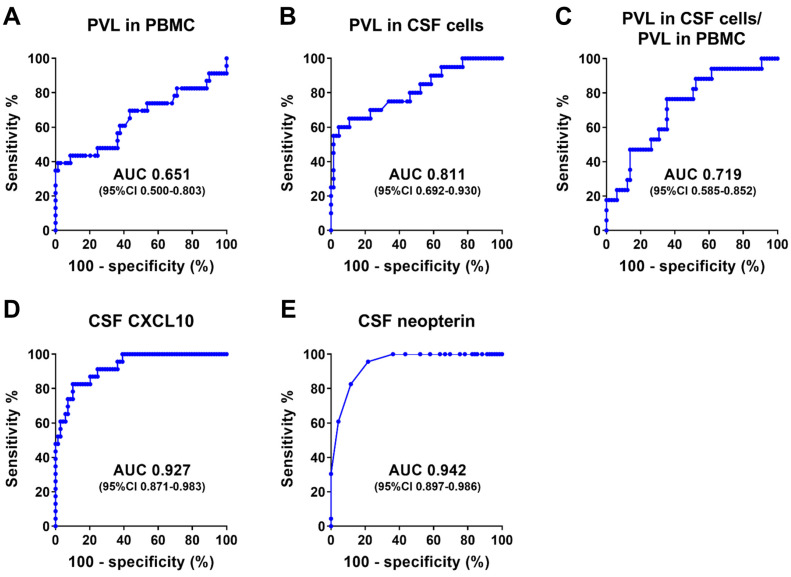
ROC analysis to assess the diagnostic performance of the five markers in discriminating patients with HAM from HTLV-1 carriers. We collected the past data of five known markers, the PVL in PBMCs, the PVL in CSF cells, the ratio of these PVLs, CSF CXCL10, and CSF neopterin, in 92 individuals (69 patients with HAM and 23 HTLV-1 carriers). PVL, HTLV-1 proviral load; PBMCs, peripheral blood mononuclear cells; AUC, area under the curve.

**Figure 5 viruses-16-01581-f005:**
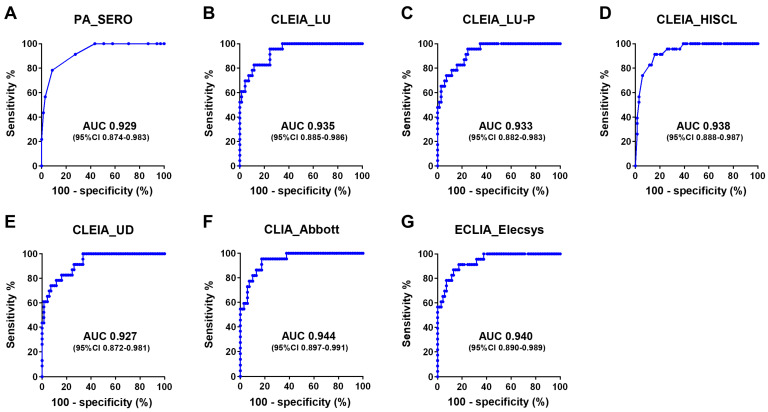
ROC analysis of CSF anti-HTLV-1 antibody levels. CSF anti-HTLV-1 antibody levels in 92 individuals (69 patients with HAM and 23 HTLV-1 carriers) were measured using seven different quantifiable anti-HTLV-1 antibody test kits, and an ROC analysis was performed for each. Information on each test kit is shown in Table 1. AUC, area under the curve.

**Figure 6 viruses-16-01581-f006:**
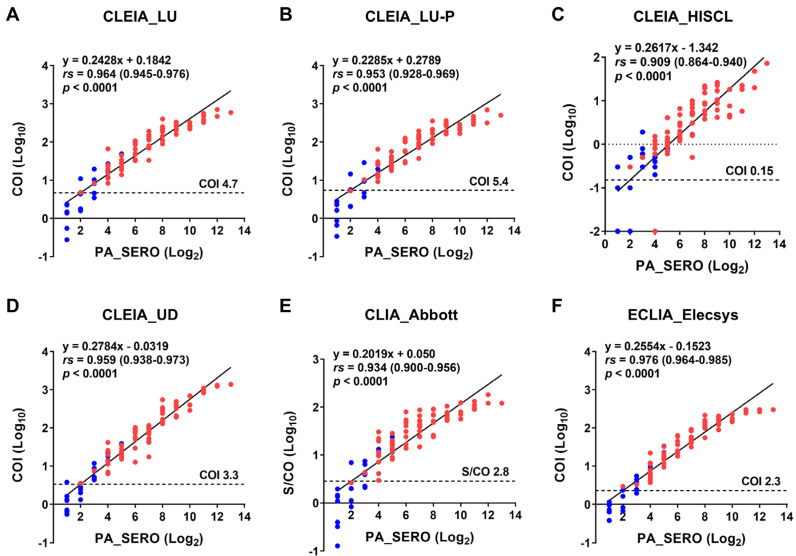
Correlations with the PA method. Using the results of anti-HTLV-1 antibody titers from 92 CSF samples (69 patients with HAM [red] and 23 carriers [blue]), we examined the correlations between the PA method and six other test kits. Antibody titers for the PA method are indicated as Log_2_ values, and antibody levels (COI and S/CO) for all other test kits are demonstrated as Log_10_ values. The linear equation, Spearman’s rank correlation coefficient (rs), 95% confidence interval, and *p*-value are shown in the figure. COI, cutoff index; S/CO, signal-to-cutoff ratio. Information on each test kit is shown in Table 1.

**Table 1 viruses-16-01581-t001:** Anti-HTLV-1 antibody test kits used in this study.

Methods	Kit’s Name	Abbrev.	Manufacturer	Ind/Sand	Cutoff Values for Blood
PA	Serodia HTLV-I	SERO	Fujirebio	n/a	Titer ≥ 16×
CLEIA	Lumipulse HTLV-I/II	LU	Fujirebio	Sand	COI ≥ 1.0
	Lumipulse Presto HTLV-I/II	LU-P	Fujirebio	Sand	COI ≥ 1.0
	HISCL HTLV-I	HISCL	Sysmex	Ind	COI ≥ 1.0
	HISCL_UD (under development)	UD	Sysmex	Sand	COI ≥ 1.0
CLIA	HTLV·Abbott (Alinity)	Abbott	Abbott Japan	Sand	S/CO ≥ 1.0
ECLIA	Elecsys HTLV-I/II	Elecsys	Roche Diagnostics	Sand	COI ≥ 1.0
IC	ESPLINE HTLV-I/II	ESPLINE	Fujirebio	Sand	*
LIA	INNO-LIA HTLV-I/II Score	INNO-LIA	Fujirebio	Ind	*

PA, particle agglutination; CLEIA, chemiluminescence enzyme immunoassay; CLIA, chemiluminescence immunoassay; ECLIA, electrochemiluminescence immunoassay; IC, immunochromatography; LIA, line immunoassay; UD, under development, Ind, indirect assay; Sand, Sandwich assay; n/a, not applicable; COI, cutoff index; S/CO, signal-to-cutoff ratio. * Based on the criteria in the manufacturer’s instructions.

**Table 2 viruses-16-01581-t002:** Characteristics of HTLV-1 carriers and patients with HAM used to determine diagnostic performance.

	HTLV-1 Carriers	Patients with HAM	*p*-Value
	n = 23	n = 69	
Sex: Female	14 (60.9%)	55 (79.7%)	0.10 ^(a)^
Age ^1^	53 [50, 69.5]	67 [60, 72]	0.02 ^(b)^
PVL in PBMCs ^1^	2.57 [0.20, 4.13]	3.52 [1.90, 5.07]	0.03 ^(b)^
PVL in CSF cells ^1^	1.46 [0.57, 4.59] ^2^	5.32 [3.89, 8.60] ^3^	<0.0001 ^(b)^
PVL in CSF cells/PVL in PBMCs ^1^	0.93 [0.63, 1.33] ^2^	1.69 [0.92, 3.20] ^3^	0.005 ^(b)^
CXCL10 in CSF (pg/mL) ^1^	249.6 [142.0, 322.7]	1090.5 [598.0, 2247.7]	<0.0001 ^(b)^
Neopterin in CSF (pmol/mL) ^1^	3 [2, 4]	8 [6, 18]	<0.0001 ^(b)^

Statistical methods: ^(a)^ Fisher’s exact test and ^(b)^ Mann–Whitney U test. ^1^ Data are expressed as median [interquartile range]. ^2^ n = 18, Data for three HTLV-1 carriers were missing, and data for the two HTLV-1 carriers that fell below the detection limit were also not included here. ^3^ n = 65, Data for four patients with HAM were missing. PVL, HTLV-1 proviral load; PBMCs, peripheral blood mononuclear cells.

**Table 3 viruses-16-01581-t003:** Overall concordance rate between the PA method with a 4× antibody titer as the cutoff value and CLEIA (Lumipulse HTLV-I/II).

		PA_SERO		Total
		Positive (≥4×)	Negative (<4×)	
CLEIA_LU	Positive (COI ≥ 1.0)	85	5	90
	Negative (COI < 1.0)	0	2	2
Total		85	7	92
Overall concordance rate				94.6%

PA, particle agglutination; CLEIA, chemiluminescence enzyme immunoassay; COI, cutoff index.

**Table 4 viruses-16-01581-t004:** Overall concordance rate between the PA method with a 16× antibody titer as the cutoff value and CLEIA (Lumipulse HTLV-I/II).

		PA_SERO		Total
		Positive (≥16×)	Negative (<16×)	
CLEIA_LU	Positive (COI ≥ 1.0)	73	17	90
	Negative (COI < 1.0)	0	2	2
Total		73	19	92
Overall concordance rate				81.5%

PA, particle agglutination; CLEIA, chemiluminescence enzyme immunoassay; COI, cutoff index.

**Table 5 viruses-16-01581-t005:** Overall concordance rate between the PA method with a 4× or a 16× antibody titer as the cutoff value and each other test kit.

	LU	LU-P	HISCL	UD	Abbott	Elecsys	ESPLINE	INNO-LIA
SERO (≥4× positive)	94.6%	95.7%	72.8%	95.7%	94.5%	96.7%	97.8%	79.3%
SERO (≥16× positive)	81.5%	82.6%	84.8%	82.6%	83.5%	88.0%	87.0%	90.2%

**Table 6 viruses-16-01581-t006:** Cutoff values and their diagnostic accuracy for each test kit for diagnosing HAM.

	SERO	LU	LU-P	HISCL	UD	Abbott	Elecsys	ESPLINE	INNO-LIA
Cutoff values for blood	Titer 16×	COI 1.0	COI 1.0	COI 1.0	COI 1.0	S/CO 1.0	COI 1.0	*	*
Sensitivity	97.1%	100%	100%	84.1%	100%	100%	100%	100%	91.3%
Specificity	73.9%	8.7%	13.0%	91.3%	13.0%	18.2%	34.8%	30.4%	65.2%
Cutoff values equivalent to the 4× titer of the PA method	Titer 4×	COI 4.7	COI 5.4	COI 0.15	COI 3.3	S/CO 2.8	COI 2.3	n/a	n/a
Sensitivity	100%	100%	98.6%	98.6%	100%	98.6%	100%	n/a	n/a
Specificity	30.4%	52.2%	47.8%	34.8%	43.5%	54.5%	43.5%	n/a	n/a
Cutoff values for maximum specificity with 100% sensitivity	Titer 4×	COI 4.7	COI 4.7	COI 0.25	COI 3.3	S/CO 2.5	COI 2.5	n/a	n/a
Sensitivity	100%	100%	100%	98.6% **	100%	100%	100%	n/a	n/a
Specificity	30.4%	52.2%	47.8%	39.1%	43.5%	54.5%	56.5%	n/a	n/a

COI, cutoff index; S/CO, signal-to-cutoff ratio; n/a, not applicable. * Based on the manufacturer’s instructions. ** Sensitivity did not reach 100% because one patient with HAM had COI = 0.

## Data Availability

The datasets generated for this study are available upon request from the corresponding author.

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
