# Peer review of "Diagnostic Value of Anti-HTLV-1-Antibody Quantification in Cerebrospinal Fluid for HTLV-1-Associated Myelopathy"

_viruses, 2024, doi:10.3390/v16101581_

Round 1
Reviewer 1 Report
Comments and Suggestions for Authors
Sato and colleagues have undertaken a comparative study to evaluate anti HTLV-1 antibodies in the CSF versus those in the sera of HAM/TSP patients and in the carriers. They have performed a correlation analysis of five different kits of immune kit with the particle agglutination (PA) method.
The methos is accurate and have tried to establish a cutoff with 100% sensitivity and maximum specificity, ranging from 43.5% to 56.5%. This cutoff, in combination with the clinical findings, will aid in the accurate diagnosis of HAM. One of the important conclusion of this paper is that the HTLV-1 antibodies can be detected in the CSF of HAM/TSP patients, in comparison to carriers.
Major comments
The study is complex considering also the limited number of patients examined, but this is a common feature when we talk about HTLV-1 infection owe to the endemic distribution and difficulty of HAM/TSP diagnosis. The study is well conducted and the conclusions are consistent with the described data. Nevertheless I would like to ask the authors the following:
1) To make more clear the vantage of sensitivity over the specificity, or vice versa, of the tested kits, the authors should in the discussion clear more broadly whether the quantitative kits are better than the qualitative for which of the two aspects.
2) The authors have to make more understandable whether the PA method is definitely less reliable that the other immune kit tested
3) The authors state that freezing and thawing twice can affect the detection of antibodies. How about one freezing and thawing. I think that it is not feasible that all the sera can be processed fresh.
4) The conclusions are rather obvious, the authors should make less obvious the sentence that the diagnosis of HAM/TSP cannot ignore the clinical features.
Comments on the Quality of English LanguageLanguage needs revision of the spelling and english form.
Author Response
Reviewer1
1) To make clearer the vantage of sensitivity over the specificity, or vice versa, of the tested kits, the authors should in the discussion clear more broadly whether the quantitative kits are better than the qualitative for which of the two aspects.
Response: Thank you for your insightful comments. According to this suggestion, we have added the following sentences to the Discussion section: (Line 387) These qualitative assays have fixed cutoffs, and their sensitivities and specificities were determined. In contrast, quantitative tests allow the cutoff point to be set freely, and the sensitivity and specificity can be changed, although there is a trade-off between them. Although we wanted to prioritize the sensitivity and set it at 100%, the sensitivity of the INNO-LIA was as low as 91.3%. Next, when the cutoff value of the quantitative test kits was adjusted to achieve a sensitivity of 91.3%, their specificities ranged from 73.9%–84.1%, which exceeded the specificity of the INNO-LIA (65.2%). This indicates that the quantitative tests demonstrated superior discriminatory ability. The other test, ESPLINA, achieved a target sensitivity of 100%; however, its specificity was low (30.4%), which was inferior to that of quantitative test kits (43.5%–56.5%).
2) The authors have to make more understandable whether the PA method is definitely less reliable that the other immune kit tested
Response: According to your suggestion, we added the following sentences: (Line 365) In addition, because the PA method involves visual determination of gelatin agglutination, there is a problem in that the interpretation of the results is more prone to variability than with other immunoassay kits that automatically measure the signal intensity of chemiluminescence or electrochemiluminescence.
3) The authors state that freezing and thawing twice can affect the detection of antibodies. How about one freezing and thawing. I think that it is not feasible that all the sera can be processed
Response: We agree that it is not feasible to process all specimens fresh. As shown in Figure 1, the results of some test kits showed a decrease in CSF antibody levels with each additional freeze-thaw cycle. Therefore, we believe that one freeze-thaw cycle is less influential than two freeze-thaw cycles. We have added the following sentence on this point: (Line 347) Considering feasibility and reproducibility, we recommend setting the number of freeze-thaw cycles to one and using the same conditions each time.
4) The conclusions are rather obvious, the authors should make less obvious the sentence that the diagnosis of HAM/TSP cannot ignore the clinical features.
Response: Thank you for your important comment. As you have pointed out, diagnosing HAM in conjunction with clinical findings is essential. To clarify this point, we have revised the Conclusion as follows (Line 430): Although this study has limitations, the cutoff values, in combination with clinical findings, would contribute to a more accurate diagnosis of HAM. Future studies are needed to identify novel markers with higher sensitivity and specificity for HAM.
Reviewer 2 Report
Comments and Suggestions for Authors
First of all, congratulations to the authors of the manuscript for the proposal presented. Not only in Japan, but in many other places, the conclusive diagnosis of HAM becomes a challenge where the development and improvement of more effective protocols in clinical evaluation and laboratory diagnosis need to be implemented. In some situations, countries with few resources and large territorial extensions have few specialized centers for the diagnosis of HTLV or HAM, whether in blood or CSF. Therefore, biological samples may take a few days to reach these diagnostic centers. Studies like this could help in this assessment. I have a few suggestions/comments that could be clarified by the authors:
1- Since this is a retrospective study, the samples had already been collected previously and were aliquoted and stored at -80ºC. How long, on average, were the samples aliquoted? Could this storage time be a crucial factor that would influence the detection of antibodies and inflammatory markers? Why did the authors not use some recently collected samples as a comparison?
2- Wouldn't 48 hours at 4ºC be too little time to assess the stability of the antibodies? Didn't the authors think of assessing within a longer period? Over 72 hours, for example?
3- Maybe I couldn't identify this information, but in what year were the 12 samples sent to the kit manufacturers obtained? The samples were collected between 2007 and 2022, correct?
4- Did the authors have information about the physical and biochemical analysis of the CSF samples? Cellularity? Biochemical markers (e.g., glucose, lactate, total proteins)? Could this somehow influence the detection of anti-HTLV-1 antibodies?
5- Regarding figure 4, if there was a significant difference between the proviral load values ​​in PBMC, CFS, in the PBMC/CSF ratio and in the inflammatory markers (CXCL10 and Neopterin) between the two groups (carriers and HAM), why do the authors use all the samples together when performing the ROC analysis? Couldn't they have analyzed the groups separately?
6- In lines 353 and 354, the authors report that the specificities were 43.5% and 56.5%. Wouldn't these values ​​be too low for these protocols to be used in the future? I understand that for diagnosis and with the option of other more specific methodologies, a test with 100% sensitivity is a good option for detecting antibodies, but couldn't this low specificity also lead to false results? What could be done to improve these specificity values?
Author Response
Reviewer 2
1) Since this is a retrospective study, the samples had already been collected previously and were aliquoted and stored at -80ºC. How long, on average, were the samples aliquoted? Could this storage time be a crucial factor that would influence the detection of antibodies and inflammatory markers? Why did the authors not use some recently collected samples as a comparison?
Response: All markers, except CSF anti-HTLV-1 antibody levels, were measured soon after CSF collection and were therefore not affected by storage time. We have revised the description in the Methods section to clarify this point (Lines 104-118). To determine CSF anti-HTLV-1 antibody levels, 92 CSF specimens were simultaneously measured at four companies in December 2022. The storage time between the CSF collection and this measurement was 5.2 ± 2.2 years (Mean ± SD). Next, to determine whether storage time at -80°C affects antibody titer, we examined the correlation between storage time and the ratio of change in antibody titer (antibody titer as of December 2022/antibody titer at diagnosis) for 92 specimens and found a significant correlation (rs = -0.2345, p = 0.0244) (See Figure below). In addition, the shorter the storage time, the higher the ratio of change in the PA method after freeze-thawing. However, when interpreting these results, we cannot rule out confounding factors or make definitive statements. Therefore, we decided to describe the limitations of this study as follows: (Line 418) The CSF specimens used to determine the cutoff values in this study were frozen and thawed twice and had long storage time (5.2 ± 2.2 years) at -80°C.
2) Wouldn't 48 hours at 4ºC be too little time to assess the stability of the antibodies? Didn't the authors think of assessing within a longer period? Over 72 hours, for example?
Response: As you have pointed out, it would have been better to add 72 h to the test conditions. However, we prioritized testing multiple kits when considering the limited amount of CSF specimens. Therefore, the study conditions were limited to three storage times, which are often clinically problematic: 0 h, 24 h, and 48 h.
3) Maybe I couldn't identify this information, but in what year were the 12 samples sent to the kit manufacturers obtained? The samples were collected between 2007 and 2022, correct?
Response: Twelve samples were collected between 2014 and 2022. The following sentence has been added to the Methods section: (Line 122) The samples were collected between 2014 and 2022.
4) Did the authors have information about the physical and biochemical analysis of the CSF samples? Cellularity? Biochemical markers (e.g., glucose, lactate, total proteins)? Could this somehow influence the detection of anti-HTLV-1 antibodies?
Response: We did not address these issues in this study because they were incomplete. For other reasons, we did not measure lactate levels in the CSF. Second, glucose levels were within the reference values in almost all patients with HAM. Third, we reported no differences in total protein and IgG levels between patients with HAM and HTLV-1 carriers (PLoS Neg Trop Dis 2013, 7(10):e2479). Lastly, in the same paper, we reported that the CSF cell count is less accurate than CSF neopterin and CXCL10 in separating HAMs from carriers. Therefore, in this study, we selected unreported findings, such as PVL in CSF cells and the ratio of PVL in CSF cells to PVL in PBMCs. To the best of our knowledge, there are no reports on the influence of these factors on the detection of anti-HTLV-1 antibodies. In light of the above, we have added the following sentence: (Line 406) Regarding other CSF markers, it is known that there is no difference between patients with HAM and HTLV-1 carriers in IgG and total protein in the CSF and that the ability of CSF cell count to distinguish HAM is lower than that of CSF CXCL10 and neopterin [29].
5) Regarding figure 4, if there was a significant difference between the proviral load values ​​in PBMC, CFS, in the PBMC/CSF ratio and in the inflammatory markers (CXCL10 and Neopterin) between the two groups (carriers and HAM), why do the authors use all the samples together when performing the ROC analysis? Couldn't they have analyzed the groups separately?
Response: We may have misunderstood your question, but because this is an ROC analysis to examine the accuracy of separating the two groups (carriers and HAM), it is necessary to use data from both groups. We do not believe that we can analyze the groups separately using ROC analysis.
6) In lines 353 and 354, the authors report that the specificities were 43.5% and 56.5%. Wouldn't these values ​​be too low for these protocols to be used in the future? I understand that for diagnosis and with the option of other more specific methodologies, a test with 100% sensitivity is a good option for detecting antibodies, but couldn't this low specificity also lead to false results? What could be done to improve these specificity values?
Response: This is an important point to be considered. As you mentioned, the CSF anti-HTLV-1 antibody test alone has limitations owing to its low specificity. Therefore, it is essential to diagnose HAM using the cutoff value of the CSF anti-HTLV antibody in conjunction with clinical findings. Additionally, to increase the specificity without reducing the sensitivity, the accuracy of the diagnostic test must be improved. This require improved detection methods and measurement of changes in the targets. However, it would be difficult to improve the anti-HTLV-1 antibody detection method further since it has already been improved to a precise “antigen sandwich assay” and a detection system with a high signal/noise ratio, such as chemiluminescence and electrochemiluminescence. Therefore, a new marker to replace CSF anti-HTLV-1 antibodies is necessary. Regarding these points, we have added the following to the conclusion (Line 430): Although this study has limitations, the cutoff values, in combination with clinical findings, would contribute to a more accurate diagnosis of HAM. Future studies are needed to identify novel markers with higher sensitivity and specificity for HAM.
Reviewer 3 Report
Comments and Suggestions for Authors
This manuscript compares the specificity and sensitivity of HTLV-1 test kits prepared by different companies using various methods. By comparing them with the standard detection methods previously used in Japan, this study identifies appropriate cutoff values to ensure these kits can be reasonably applied in clinical testing.
Line 106: “Genomic DNA”, HTLV-1 is an enveloped single-stranded RNA retrovirus, so here should be Genomic RNA.
Line 131: This is a suggestion: the manuscript should clarify the definitions of "HAM patients" and "HTLV-1 carriers."
Line 164: “to decrease after….” Not all Figures 1B-1F show a decreasing trend, Figures D and F do not follow this pattern. Therefore, recommend the manuscript provide a more accurate description of Figures 1B-1F.
Line 166: “showed an increasing trend…” In fact, there is no significant increase trend.
Line 295: “other test kits.” Here only describes kits IC_ESPLINE and LIA_INNO-LIA, so using the term "other test kits" may confuse readers.
Most of the discussion section in this manuscript focuses on restating the results, which might give readers a sense of redundancy. It would be more effective if the manuscript could include comparisons with other methods or studies to highlight the strengths of this research.
Author Response
Reviewer 3
Line 106: “Genomic DNA”, HTLV-1 is an enveloped single-stranded RNA retrovirus, so here should be Genomic RNA.
Response: To measure the copy number of HTLV-1 provirus integrated into lymphocytes, we used “Genomic DNA,” not “Genomic RNA.” We think it was difficult to understand why we used genomic DNA, so we have added the following description: (Line 114) To measure the proviral load in CSF cells, genomic DNA was extracted using the phenol-chloroform method.
Line 131: This is a suggestion: the manuscript should clarify the definitions of "HAM patients" and "HTLV-1 carriers."
Response: Following the proper advice, we added the definitions of “HAM patients” and “HTLV-1 carriers”: (Line 102) Enrolled patients with HAM were diagnosed according to the WHO diagnostic criteria [18], and those not diagnosed were defined as HTLV-1 carriers.
Line 164: “to decrease after….” Not all Figures 1B-1F show a decreasing trend, Figures D and F do not follow this pattern. Therefore, recommend the manuscript provide a more accurate description of Figures 1B-1F.
Response: Since the description of the results was not accurate, we revised it as follows: (Line 167) In contrast, for the other methods (Figures 1B-1F), the antibody levels (COI and S/CO) after the third freeze-thaw cycle were lower than those after the first freeze-thaw cycle. For three of the test kits (LU, LU-P, and Abbott), the antibody levels decreased significantly with each freeze-thaw cycle (Figures 1B, 1C, and 1E).
Line 166: “showed an increasing trend…” In fact, there is no significant increase trend.
Response: Thank you for pointing this out. We have rewritten this as follows: (Line 171) the PA method (SERO) showed an increasing trend in antibody titers as storage time increased from 0 to 48 h (Figure 2A). However, changes in antibody levels over the 48-hour storage period in all methods, including the PA method, were not significant (Figures 2B-2F).
Line 295: “other test kits.” Here only describes kits IC_ESPLINE and LIA_INNO-LIA, so using the term "other test kits" may confuse readers.
Response: We have corrected the original sentence as follows: (Line 301) The LIA showed a sensitivity of 91.3% and a specificity of 65.2%, which was higher in specificity but lower in sensitivity than that of the quantifiable test kits (lower part of Table 4).
Most of the discussion section in this manuscript focuses on restating the results, which might give readers a sense of redundancy. It would be more effective if the manuscript could include comparisons with other methods or studies to highlight the strengths of this research.
Response: We agree with the reviewer’s comments. First, we deleted the following sentence, which merely reiterates the results (Line 328 in the original manuscript): No significant difference was observed in CSF antibody titers between the steroid-treated and steroid-free groups (p = 0.627). Next, we collected and reviewed articles on the detection of CSF anti-HTLV-1 antibodies in HAM patients and HTLV-1 carriers and added the following paragraph to the Discussion section (Line 325): To date, various methods have been used to detect anti-HTLV-1 antibodies in the CSF of patients with HAM and HTLV-1 carriers. First, it was reported that PA (Serodia-ATLA) and recombinant gag-env hybrid protein-coated ELISA are useful for detecting anti-HTLV-1 antibodies in the CSF and diagnosing HAM [20]. Western blotting has detected GD21, rgp46-I, and p24 as targets of CSF anti-HTLV-1 antibodies in many patients with HAM [21]. In addition, an EIA using 16 synthetic peptides derived from HTLV-1 gag and env was used to measure the peptide-specific IgG antibody levels. Patients with HAM show an intrathecal immune response to more gag and env epitopes than HTLV-1 carriers do [22]. However, in these studies, the accuracy of HAM diagnosis was not examined. In contrast, a study by Kodama et al. showed the diagnostic accuracy of CSF anti-HTLV-1 antibody when using the CLIA and CLEIA methods [10]. However, it showed the cutoff and diagnostic accuracy when the PA method was considered true and did not show the accuracy of diagnosing HAM clinically. In this regard, this study clarified the diagnostic accuracy of HAM using additional test kits.
Round 2
Reviewer 3 Report
Comments and Suggestions for Authors
The studies in this manuscript identified a cutoff value applicable to multiple HTLV-1 antibody detection kits that, when combined with clinical findings, will aid in the accurate diagnosis of HAM.